# Mesenchymal Stem Cell-Derived Secretome: A Potential Therapeutic Option for Autoimmune and Immune-Mediated Inflammatory Diseases

**DOI:** 10.3390/cells11152300

**Published:** 2022-07-26

**Authors:** Györgyi Műzes, Ferenc Sipos

**Affiliations:** Immunology Division, Department of Internal Medicine and Hematology, Semmelweis University, Szentkirályi Street 46, 1088 Budapest, Hungary; dr.siposf@gmail.com

**Keywords:** mesenchymal stem cell-derived secretome, extracellular vesicles, exosomes, autoimmune diseases, immune-mediated inflammatory diseases

## Abstract

Immune-mediated inflammatory diseases (IMIDs) encompass several entities such as “classic” autoimmune disorders or immune-mediated diseases with autoinflammatory characteristics. Adult stem cells including mesenchymal stem cells (MSCs) are by far the most commonly used type in clinical practice. However, due to the possible side effects of MSC-based treatments, there is an increase in interest in the MSC-secretome (containing large extracellular vesicles, microvesicles, and exosomes) as an alternative therapeutic option in IMIDs. A wide spectrum of MSC-secretome-related biological activities has been proven thus far including anti-inflammatory, anti-apoptotic, and immunomodulatory properties. In comparison with MSCs, the secretome is less immunogenic but exerts similar biological actions, so it can be considered as an ideal cell-free therapeutic alternative. Additionally, since the composition of the MSC-secretome can be engineered, for a future perspective, it could also be viewed as part of a potential delivery system within nanomedicine, allowing us to specifically target dysfunctional cells or tissues. Although many encouraging results from pre-clinical studies have recently been obtained that strongly support the application of the MSC-secretome in IMIDs, human studies with MSC-secretome administration are still in their infancy. This article reviews the immunomodulatory effects of the MSC-secretome in IMIDs and provides insight into the interpretation of its beneficial biological actions.

## 1. Introduction

The term immune-mediated inflammatory diseases (IMIDs) encompasses several disease entities including definite, “classic” autoimmune disorders (ADs) (among others as systemic lupus erythematous/SLE/, rheumatoid arthritis/RA/); neurological disorders (such as multiple sclerosis); cutaneous inflammatory conditions (like psoriasis and atopic dermatitis); immune-mediated disorders with autoinflammatory characteristics such as the spondyloarthritis (SpA) disease spectrum, inflammatory bowel disease (IBD), asthma, graft-versus-host disease as well as other conditions in which the immune system is involved, but not the primary element of the pathogenesis (e.g., cancer or infectious disorders) [1,2].

Approximately 3–5 percent of the world’s population is currently affected by ADs, of which there are over 100 different types, and both rates are still rising [3]. Within the multifactorial etiology of chronic, self-perpetuating ADs with prominent inflammation, failure of self-tolerance, and abnormal immune regulation appear to be particularly harmful. Thus, the primary therapeutic objective in ADs is to achieve immune homeostasis, which is the dynamic state of equilibrium between pro- and anti-inflammatory responses. Although treatment strategies have improved as a result of a greater understanding of the pathogenesis of ADs, immunosuppressive drugs have failed to prevent relapses in some patients (primarily those with refractory diseases). In addition, harmful side effects such as an increased risk of infection and a heightened susceptibility to tumor development, may occur in tandem with long-term administration. As part of personalized medicine, it is necessary to develop more specific and tolerable treatment options for ADs. 

According to their plasticity, stem cells can be categorized as totipotent, which give rise to all embryonic and extraembryonic cell lines; pluripotent, which can produce all embryonic cell types; multipotent, which differentiate into a large number of cell types; oligopotent, which can differentiate into only a few cell lineages; and unipotent, which give rise to only one specific cell type [4,5]. Multipotent adult stem cells (SCs) such as hematopoietic stem cells and mesenchymal stem cells (MSCs) are by far the most commonly used type in clinical practice because they are readily available from patients with a variety of medical conditions. The majority of SCs are lineage-restricted and refer to their tissue of origin. Endothelial, mesenchymal, and adipose tissue-derived cells can be distinguished [4,5,6]. In addition to bone marrow, a variety of other tissues can be used to isolate mesenchymal stem cells (adipose tissue, peripheral blood, placenta, dental pulp, synovial membrane, periodontal ligaments, endometrial, trabecular, and compact bones). They can develop into mesodermal, endodermal, and ectodermal cells under the proper culture conditions [4,5,6]. Allogeneic or autologous SC therapy has been viewed as a promising alternative treatment over the past two decades [7]. In addition to repairing damaged tissues, SCs have the unique ability to modulate the immune system extensively in terms of immune reconstitution and overcoming the impaired self-tolerance status. According to these activities, SCs could serve as ideal therapeutic agents with the capacity to provide long-lasting, sustained protection against autoimmunity [7]. However, the majority of SC treatments for ADs in humans are still in their infancy, necessitating additional research prior to their anticipated widespread clinical application. On the other hand, however, SCs have a number of disadvantages including host cell rejection, ectopic tissue formation, detrimental effects on the pulmonary microvasculature, or protumor activities [8]. Because of these obstacles, their clinical application has still been limited. Thus, additional, more harmless MSC-based therapeutic strategies are required.

Due to the possible side effects of MSC-based treatments, there has been an increase in interest for the secretome produced by MSCs as an alternative therapeutic application. With the utilization of the secretome of MSCs, a number of major, partly technical advantages may become available including no requirement for intrusive solution to extract cells, the capacity to undertake pharmacological dose and safety testing, the convenience of application, and the ability to successfully manipulate its composition. In addition, secretome derived from MSCs plays a prominent role in the regulation of innate and adaptive immune responses [9]. 

The use of MSC-secretome as a cell-free therapy is being studied in a number of disease groups in experimental models and clinical trials. Therapeutic effects were found to be promising in skeletal muscle regeneration (acute and chronic muscle injuries, atrophic muscle disorders) [10], traumatic and degenerative intervertebral disc alterations [11], dermatological disorders (e.g., wound healing, photoprotection, hair growth, alopecia, antimicrobial effects on skin wounds) [12], liver diseases (such as liver failures, partial hepatectomy) [13], cardiovascular diseases [14], neurological degenerative diseases (e.g., Alzheimer’s and Parkinson’s disease) [14], bone diseases [14], or osteoarthritis [15]. MSC-derived secretome displays a dual function in tumor promotion and tumor suppression [16]. Moreover, MSC-derived secretome also displays a context-dependent dual role in the aging of other MSCs [16]. In addition, the MSC-secretome can cause cell and organ injuries such as microbial ischemic brain injury, the calcification of smooth muscle cells, degeneration of skeletal muscle, chronic and acute renal injury, pancreatic degeneration, and acute respiratory distress/injury [17]. The purpose of this review is to summarize the current trends in MSC-secretome therapy options for ADs and other IMIDs from the perspectives of both basic research and clinical practice.

## 2. Organization and Isolation of the MSC-Secretome

Soluble and vesicular factors derived from MSCs (and other cells) exhibit a variety of unique properties that can make them a precious tool for therapeutic reasons [18]. The soluble fraction is abundant in immunomodulatory molecules, cytokines, chemokines, and growth factors. The vesicular fraction consists of extracellular vesicles (EVs) [18], which are mainly categorized based on their size [19,20,21].

*Exosomes*, also known as *nanovesicles*, correspond to small EVs. They originate from the endocytic pathway and range in size from 30 to 200 nm on average, and express CD63, CD9, CD81, tumor susceptibility 101, flotillin, Alix, and the endosomal sorting complex required for transport-3 as common markers. Typically, exosomes are composed of secondary metabolites, proteins, nucleic acids (such as mRNA, miRNA, and other non-coding RNAs), and lipids [19,20,21]. The *microvesicles (ectosomes)* represent EVs of medium size. These EVs originate from the plasma membrane of the cell with an average size of 200 to 1000 nm. Their common surface markers are integrins, selectins, CD40 ligand, ADP-ribosylation factor (ARF) 6, and vesicle-associated membrane protein 3, and typically contain secondary metabolites, lipids, proteins, and nucleic acids, such as mRNA, miRNA, and other non-coding RNAs [19,20,21]. The *oncosomes* and *apoptotic bodies* are examples of *large EVs*. The term “oncosomes” refers to the simple fact that these EVs originate from cancer cells [22]. These large EVs may serve as a source of cancer-specific biomarkers extractable from biological fluids [22]. Oncosomes are larger than 1000 nm in size, and ARF2, annexin A1, and annexin A2 are their most prevalent surface markers. Apoptotic bodies are between 50 and 100 μm in diameter, and are released by dying cells. Annexin V, phosphatidylserine, and thrombospondin are their most common markers. Large EVs contain nuclear fractions, organelles, proteins, DNA, coding and non-coding RNA, and lipids. All EVs participate in intercellular communication, with the exception of apoptotic bodies, which typically function in phagocytosis (Figure 1) [19,20,21,22].

Due to the highly complex nature of the secretome, isolating products derived from the MSC-secretome is a significant issue in biotechnological development. The type and quantity of secreted products depend on the culture period or growth phase of the culture from which the secretome is separated [23]. The isolation technique of particular products from the medium must also be taken into account [23]. Existing techniques for isolating EVs face numerous obstacles [24,25]. Ultracentrifugation, which is currently regarded as the gold standard for separating these structures and is capable of handling relatively large volumes, has recovery rates as low as 25 percent and is a lengthy, tedious process [26,27]. In addition, there is evidence that the high g forces (usually 100,000 g) can influence the functional properties of the EVs themselves [26]. Size exclusion and antibody-based capture mechanisms have low throughput and require a final concentration step [28]. Moreover, the high cost of antibody-based mechanisms restricts their use in numerous applications. Microfluidic techniques permit the simultaneous isolation and characterization of EVs, but the recovery of EVs from this equipment remains difficult [25]. In recent years, anion-exchange chromatography has been used to extract EVs with high purity in a single step, taking advantage of their negative charge [29]. Clinical applications have a great need for EV isolation technologies that allow for high throughput, preserve EV integrity and biological properties, and are not prohibitively expensive. 

## 3. Immunomodulatory Effects and Manipulation of the MSC-Secretome

The primary mechanism of the immunoregulatory actions of MSCs corresponds to intercellular interactions and paracrine signaling. The MSC-secretome is the principal route by which cells induce paracrine actions. Various constituents of the MSC-secretome have been demonstrated to regulate cells of both innate and adaptive immunity upon the delivery of DNA, mRNA, circular RNAs (circRNAs), and long non-coding RNAs (lncRNAs) [30,31]. 

The effect of the MSC-secretome on innate immune cells acts against mechanisms of immune dysregulation. MSC-exosomes improve the anti-inflammatory phenotype of human regulatory macrophages by downregulating IL-22 and IL-23, thereby promoting inflammation remission [32]. Additionally, they stimulate the differentiation of macrophages toward M2 via the miR-223/pKNOX1 (PBX/knotted 1 homeobox 1) pathway [33]. Through the TLR4/nuclear factor (NF)kB/phosphatidylinositol 3 kinase (PI3K)/Akt signaling cascade, extracellular vesicles derived from MSCs also promote polarization toward the M2 phenotype in neonatal mice [34]. In rabbits, transplanted MSCs underwent extensive apoptosis shortly after transplantation [35]. Apoptosis plays a crucial role in the activation of the inflammatory regulatory capabilities of MSCs [36]. Another study using a murine model of GvHD demonstrated that after transplantation, MSCs underwent extensive caspase activation and apoptosis, which is necessary for their immunosuppressive function. It was discovered that apoptotic bodies derived from MSCs induce macrophage polarization toward the M2 phenotype [37]. Moreover, MSC-exosomes derived from fetal liver were shown to inhibit NK cell proliferation, activation, and cytotoxicity via TGFβ regulation [38].

Regarding antigen presentation and adaptive immunity, MSC-secretome components definitely contribute to tolerogenicity. EVs derived from human bone marrow MSCs inhibit dendritic cell maturation and function by expressing anti-inflammatory IL-10 and TGFβ, and decrease the production of pro-inflammatory IL-6 and IL-12p70 [39]. Furthermore, by inducing murine tolerogenic dendritic cells with a low expression of co-stimulatory markers, MSC-exosomes promote the differentiation of regulatory T cells by increasing IL-10 and TGFβ expression and decreasing IL-6 production [40]. Human MSC-EVs enhance T cell apoptosis and inhibit T cell proliferation by increasing the expression of IL-10 and TGFβ [41]. According to mice experiments, MSC-exosomes could also facilitate T cell proliferation and influence the cell cycle via the p27kip1/CDK2 pathway [42]. Similarly, in mice, they suppress Th1 differentiation by encapsulating miR-23a-3p and post-transcriptionally controlling TGFβ receptor 2 [43]. Upon targeting the PI3K/Akt signaling pathway, human MSC-EVs can also impede B cell activation (Figure 2) [44].

As MSCs have a high degree of immunoplasticity, the composition of their secretome varies depending on their context. MSCs sense biomolecules in their environment partly through Toll-like receptors (TLRs). TLR4 activation by stimuli like lipopolysaccharides (LPS) triggers the MSC1 phenotype, and so cells primarily produce proinflammatory mediators. Conversely, factors such as tumor necrosis factor-α (TNFα), interferon-γ (IFNγ), or poly-I:C result in TLR3 induction and MSC polarization toward an MSC2 phenotype, resulting in the production of mediators that control and manage the function of immune effector cells, and consequently contributing to tissue repair and immune homeostasis [45]. The possibility of manipulating the microenvironment of MSCs to polarize them toward an MSC2 phenotype that produces a secretome rich in anti-inflammatory factors seems to currently be fashionable. 

The MSC-secretome can be manipulated in various ways. MSCs are capable of secreting more immunomodulatory factors under hypoxic culture conditions, with oxygen concentration ranging from 0 to 10% when using a cell-culture chamber or bioreactors [45,46,47]. Undoubtedly, hypoxic preconditioning can boost the MSC production of pleiotropic growth factors such as hepatocyte growth factor (HGF), insulin-like growth factor-1 (IGF-1), or brain-derived neurotrophic factor, and further immunomodulatory factors such as indoleamine-2,3-dioxygenase (IDO), interleukin-10 (IL-10), or prostaglandin E2 (PGE2) [48,49,50,51,52].

Recent research indicates that when hypoxic preconditioning is utilized, the EV density and cargo could also be altered. However, hypoxia does not affect the average size, morphological appearance, or surface markers of MSC-derived EVs [53,54]. Despite reports indicating that that the release of EVs can be triggered [55] by MSCs cultured under hypoxic conditions, the real situation is still uncertain [54,55,56]. The discrepancy may originate from the degree of hypoxia, since even small alterations in oxygen concentration and exposure time can exert significant changes [55]. Furthermore, it is important to note that while some studies have shown that hypoxia may expand cellular lifespan [57], in contrast, others have indicated the death of cells [58].

MSCs produce more immunomodulatory factors in three-dimensional (3D) cultures [59]. For a MSC 3D culture, spheroid formation is the most accepted strategy [60,61]. In this situation, less oxygen diffuses to the inner layer of cells, and the created hypoxic microenvironment not just promotes cell–cell contacts, but alters the immunomodulatory factor secretion (e.g., increasing granulocyte colony-stimulating factor /G-CSF/, IL-1a, IL-10, PGE2, or decreasing TNFα levels) [60,62,63,64]. Furthermore, the cell culture on scaffolds also has a direct effect on secretome composition and function by increasing the HGF levels [65]. Nonetheless, one of the most intriguing methods for obtaining a 3D-MSC-secretome is the encapsulation of MSCs in hydrogels [66,67]. Hydrogels make it easy to change the mechanical properties of the proximate cell microenvironment (such as elasticity and stiffness) and provide it with patterns that can be found in the natural extracellular matrix, which all contribute to enrich the secretome [68].

The method of licensing (i.e., the exposure of MSCs to biochemical stimuli) has been extensively investigated. TLR3 upregulation by inflammatory cytokines such as IFNγ, TNFα or IL-1 may contribute to the polarization of MSCs toward the MSC2 anti-inflammatory phenotype, producing HGF, galectin-9 (Gal-9), IDO, PGE2, transforming growth factor β (TGFβ), or cyclooxygenase 2 (COX2) [69,70,71]. Exposing MSCs to these bioactive molecules has become a common approach in therapies based on the MSC-derived secretome (Figure 3) [72,73].

## 4. Therapeutic Applications of the MSC-Secretome for Autoimmune and Immune-mediated Inflammatory Diseases

### 4.1. Systemic Lupus Erythematosus

Systemic lupus erythematous (SLE), an idiopathic, multifactorial chronic autoimmune disease, is defined mainly by immune dysregulation, antibodies to nuclear and cytoplasmic antigens, systemic inflammation with a wide spectrum of clinical manifestations, and a relapsing-remitting course [74].

Although the use of autologous MSCs in autoimmune and inflammatory diseases appears promising [75,76], there is evidence for obvious MSC deficiency in SLE [77]. MSCs derived from SLE patients exhibit a variety of structural and functional failures compared to healthy MSCs [78,79]. SLE-originated MSCs display abnormalities in their growth, proliferation, differentiation, apoptosis, migration, cytokine production, and immunomodulatory properties. The theoretical basis for using the MSC-secretome in SLE is partly due to the MSC limitations and partly because of the fact that the secretome could affect almost all types of immunocompetent cells.

In SLE, the effect of the MSC-secretome on innate and adaptive immune cells acts against the mechanisms of immune dysregulation. Cargoes of extracellular vesicles such as DNAs, mRNAs, miRNAs, lncRNAs, and circRNAs have been shown to regulate autoimmunity and are associated with SLE activity [80,81]. MSC-exosomes from the human umbilical cord were found to increase the fraction of M2 macrophage polarization both in murine and human lupus [82,83]. Moreover, a recent study confirmed that exosomal tsRNA-21109 from MSCs alleviates SLE by inhibiting macrophage M1 polarization [84]. Human MSC-EVs inhibit T cell differentiation toward Th1 via glycolysis and cytokine pathway regulation [85]. 

Accumulating evidence suggests that non-coding RNAs, miRNAs, and circRNAs derived from MSC-EV play a crucial role in the pathogenesis of SLE [31,80,86]. Despite the fact that MSC therapies have demonstrated promise in models of SLE-related pathologies, the function and mechanism of MSC-exos remain unknown. However, it is clear that the MSC-secretome possesses potent anti-inflammatory and immunomodulatory effects [31]. Therefore, the application of the MSC-secretome could be considered as an ideal therapeutic option for severe, refractory SLE. Additionally, MSC-secretome may represent a cell-free, low-immunogenicity alternate solution to disadvantaged MSC therapies [31].

### 4.2. Inflammatory Arthritides

Rheumatoid arthritis (RA) is a systemic autoimmune disease of unknown origin that is primarily characterized by chronic, progressive polyarthritis, but also frequently involves a broad spectrum of extra-articular organs by terms of inflammation [87]. In RA, patients lose self-tolerance, and start to produce autoantibodies.

There are numerous conventional and novel disease-modifying anti-rheumatic drugs available for the treatment of RA. Along with their immunosuppressive and immunomodulatory effects, however, several adverse effects must also be considered [87]. Thus, the secretome derived from MSCs with immunomodulatory properties could serve as a suitable therapeutic option.

Many intriguing results have already been obtained from animal experiments within that direction. Not only was the secretome from MSC found to have a significant anti-inflammatory effect in a murine model of collagen-induced arthritis (CIA), but this effect varied among the different secretome fractions [88]. In this experiment, the MSC-derived exosomes were more effective than the microvesicles in increasing the number of regulatory T cells. Additionally, MSC-derived microvesicles induced significantly less IL-10 and TGFβ production by the T and B cells than MSCs per se [89].

Proliferation, invasion, and inflammatory response abnormalities of rheumatoid fibroblast-like synoviocytes (RA-FLS) are critical in the progression of RA. Exosomes derived from modified MSCs (Exo-150; containing miR-150-5p) were found to target matrix metalloproteinase 14 (MMP14), and thus inhibit the migration and invasion of RA-FLS [90]. Exo-150 injections decreased the hind paw thickness and clinical arthritic scores in mice with CIA. Exo-150 also inhibited synoviocyte hyperplasia and angiogenesis, thereby reducing joint destruction.

Human umbilical cord MSC-derived exosomes (hUCMSCEs) have also been shown to beneficially influence bone damage in CIA rats [91]. The exosomes inhibited bone destruction by rebalancing the receptor activator of the NFκB ligand/osteoprotegerin (RANKL/OPG) ratio in a concentration-dependent manner. Furthermore, the intra-articular administration of hUCMSCEs markedly suppressed the serum and synovial fluid levels of the C–C motif chemokine ligand 2 (CCL2) and C-X-C motif chemokine ligand 12 (CXCL12) in the experimental animals [92]. 

Exosomal circRNAs derived from MSCs could also be a promising therapeutic approach for RA treatment. MSC-derived exosomal circFBXW7 treatment inhibited RA-FLS proliferation, migration, and inflammatory response as well as the RA rat model damage by sponging miR-216a-3p and activating histone deacetylase 4 (HDAC4) (Figure 4) [93].

Although animal experiments have been extremely encouraging and no significant adverse effects associated with the use of secretome have been reported, human trials have not begun yet. It should also be borne in mind that some new research [94,95] suggests that epigenetically modified bone-marrow-derived MSCs (and presumably also their secretome) producing more IL-6 may promote the progression of RA by migrating to the synovium. In any case, more research is required to investigate MSC-based and cell-free therapeutic options for human RA.

Spondyloarthritis (SpA) is defined by synovial and entheseal involvement, resulting in spinal and oligoarticular peripheral arthritis, most commonly in genetically predisposed (HLA-B27+) individuals. Additionally, extra-musculoskeletal, systemic inflammatory organ manifestations may also occur. SpA may be listed as an autoinflammatory disease, with a predominant involvement of innate immunity. Adult spondyloarthritis can be classified into several well-defined subtypes; (one of the most known is Bechterew’s disease) [96]. 

Interestingly, the use of MSCs and MSC-secretome in the SpA group can cause not only beneficial but also detrimental effects. In mice with a low expression of A20 (a modulator of signal transducer and activator of transcription 1/STAT1/-dependent gene transcription) in myeloid cells, early enthesitis has been found [97]. In parallel, the inhibition of the JAK/STAT1 pathway resulted in clinical and histological improvement of enthesitis [97]. It would not be unexpected if a similar defect in human BM-MSCs contributed to the development of enthesitis in response to TNF stimulation. In a TNF transgenic mouse model, the selective expression of TNF receptor I in mesenchymal cells was detected, and enthesitis occurred regardless of the presence or absence of mature T and B cells. This indicates that only mesenchymal cells are crucial for the development of joint inflammation [98,99]. Because reactive arthritis sometimes precedes spondyloarthritis, it is hypothesized that transient or persistent silent infections affecting the bone marrow may contribute to the adverse phenotyping of bone marrow-derived MSCs [100]. Likewise, the role of EVs has been found to promote the initiation and spread of the underlying disease (including spondyloarthritides), especially in obese people, through an interaction between adipocytes and immune cells mediated by EVs [101,102]. In spondyloarthritides, the study of positive or negative therapeutic effects of the MSC-secretome is essential. In addition to animal models and cell culture experiments, detailed human studies must be introduced.

### 4.3. Multiple Sclerosis

Multiple sclerosis (MS) is an inflammatory demyelinating autoimmune disease of the central nervous system in which the protective myelinated axons of the nerve fibers in the brain and spinal cord are targeted, along with parallel disruption of the blood–brain barrier. MS can be present in several forms such as a progressive or a fluctuating, relapsing-remitting type [103,104,105].

The success of using MSCs in MS has fallen short of the expectations. In patients with the relapsing-remitting type, disease-modifying treatment altered the activity of MSCs [106]. Additionally, it was established that the patients’ own MSCs were unsuitable for disease prevention [107]. MSC differentiation into neurons, however, has a significant neuroprotective effect [108]. MSC-exosomes, as nanoscale vesicles coated with a lipid membrane, readily cross the blood–brain barrier. Exosomes can be used to modify autoreactive T cells in a pro-tolerance direction and deliver factors to the central nervous system that promote neuronal survival and remyelination [108]. Thus, their utility in MS is almost self-evident.

By modifying exosomes, receptors may be added to their surface, allowing for targeted drug delivery to cells. Exosome-encapsulated curcumin or JSI124 (a selective STAT3 inhibitor) were delivered to microglia cells via an intranasal route in mice [109]. The results of the experimental autoimmune encephalitis (EAE) demonstrated that mice treated intranasally with modified exosomes are protected from the progression of the inflammatory disorder induced by the myelin oligodendrocyte glycoprotein peptide. Intranasal administration of the modified exosomes resulted in the rapid delivery of the exosome-encapsulated drug to the brain, where it was selectively taken up by microglial cells and subsequently induced their apoptosis [109]. MSC-derived EVs also have an effect on splenic mononuclear cells in EAE mice, increasing their IL-10 secretion. Moreover, they can exert immunoregulatory effects by delivering potent biosubstances to immunocompetent cells via EVs (e.g., TGFβ, programmed death-ligand 1, Gal-1) [110]. These findings may help to explain the efficacy of the MSC-secretome in the EAE mice model. Additionally, in a mouse model of progressive MS (i.e., Theiler’s murine encephalomyelitis virus /TMEV/-induced demyelinating disease) it was demonstrated that EVs exert potent immunoregulatory actions. IV administration of EVs derived from human adipose-derived MSCs ameliorated motor deficits in infected mice via immunomodulatory effects, reducing brain atrophy and promoting remyelination [111]. In contrast, however, it was demonstrated that exosomes secreted by microglia during early TMEV infection contained viral RNA and were capable of activating uninfected bystander CNS cells, thereby promoting an inflammatory immune response. Thus, exosomes secreted by microglia during virus infection could contribute to viral persistence and neuroinflammation, both of which contribute to the development of demyelinating disease [112].

The therapeutic efficacy of MSC-exosomes is also supported by the fact that MSC-derived exosomes injected into the tail vein of EAE rats exhibited immunoregulatory capacity, indicating that exosomes from MSCs could regulate microglia polarization from the M1 to M2 phenotype [113]. Additional cell-specific therapies are also possible via exosome manipulation. Recently, covalent conjugation of the carboxylic acid-functionalized LJM-3064 aptamer to the amine groups on the MSC-exosome surface was performed. The aptamer-exosome bioconjugate was found to promote the oligodendroglia cell line proliferation in vitro. Furthermore, upon in vivo administration of the prepared bioconjugate to female EAE-mice, the inflammatory response was suppressed along with a decrease in the demyelinating CNS lesions, thus resulting in a reduction in the disease severity [114].

Exosomes loaded selectively and made their membrane cell specific enables the development of additional targeted treatment options. Exosomes isolated from rhesus monkey MSCs were utilized in two mouse models of demyelination (EAE and a toxic demyelination model using cuprizone; CPZ) [115]. In the CPZ model, MSC-Exos significantly improved the neurological outcome, decreased the amyloid precursor protein density, decreased neuroinflammation by increasing the M2 phenotype of microglia and their associated cytokines, and inhibited the TLR2/interleukin 1 receptor associated kinase 1/NFκB pathway. Moreover, the MSC-Exo treatment considerably promoted cognitive function, facilitated remyelination, increased M2 phenotypic polarization, and inhibited TLR2 signaling [115].

Exosomes derived from neural stem cells were targeted using a lentivirus-armed PDGFR-alpha ligand capable of anchoring the membrane [116]. Exosomes modified in this manner appeared to be capable of targeting oligodendrocyte progenitor cells in the neural lesion area. Following this, the exosomes were loaded with bryostatin-1, and the targeted exosomes were administered to the CPZ-treated mice. The data demonstrated that loaded-armed exosomes have a potent therapeutic effect. This novel exosome-based delivery method significantly enhanced the myelin sheath’s ability to protect integrity and promote remyelination. Additionally, it inhibited astrogliosis and axon damage as well as pro-inflammatory microglia [116].

Although these procedures may represent a paradigm shift in the treatment of MS, their clinical utility remains unknown. In a recent study, the suppressive activity of hUCMSCEs was investigated in vitro in co-culture with peripheral blood mononuclear cells (PBMCs) [117]. It was demonstrated that MSCs inhibited PBMC proliferation and that MSC-derived exosomes were more effective at suppressing proliferation with a low alloreaction rate. However, after extensive sub-culturing, the suppression capacity of MSCs and their exosomes decreased significantly [117]. Regrettably, additional evaluations are necessary to progress toward a clinically applicable functional approach. According to www.clinicaltrials.gov (accessed on 22 May 2022), only one study is planned in relapsing MS, in which, as a sub-target, the analysis of molecular and biochemical parameters including circulating exosomes are planned to perform and compare responder and non-responder patients.

### 4.4. Uveitis

Uveitis is an intraocular inflammatory condition, a form of potentially blinding diseases that can affect either one or both eyes. Different kinds of uveitis may affect various areas of the eye [118]. Uveitis can arise as a result of the immune system countering an eye infection, but it can also happen as a consequence of the immune system, attacking healthy tissue in the eyes, and some forms are considered due to aberrant autoimmune responses [118]. Conventionally, systemic and local methods have been used to treat the disease, though their long-term side effects are frequently severe. 

Animal models of experimental autoimmune uveitis are frequently used to study the disease [119]. Upon a rat model of experimental autoimmune uveitis (EAU) [120], it has been demonstrated that the topical application of MSC-Exos can inhibit the autoimmune response in vivo. It was discovered that periocular injections of human MSC-Exo can decrease immune cell infiltration, preserve retinal structure, and restore retinal function in EAU rats. This study suggests that MSC-Exos may be useful in treating autoimmune uveitis in humans. Significantly, xenogeneic MSC-Exo from humans improved the autoimmune reaction in rats [120].

The post-uveitis macular hole is a potentially severe complication [121]. In a recent pilot study, the efficacy of the intravitreal injection of MSC-Exos following standard pars plana vitrectomy (PPV) in conjunction with inverted internal limiting membrane (ILM) flap technique and ILM peeling was evaluated for the treatment of massive and resistant macular holes [122]. The findings suggest that intravitreal injection of MSC-Exos at the end of standard PPV may improve the anatomical and visual outcomes of surgery for macular holes that have been resistant to treatment. MSC-Exo therapy was found to be safer than MSC therapy in terms of the cell proliferation risk, and it was less invasive since no additional surgery was required [122]. In posterior uveitis, cells of the retinal pigment epithelium (RPE) may be destroyed by inflammatory activities [122]. RPE cells may release immunosuppressive exosomes that suppress Th17 and Th22 cells [123]. In light of the immunoregulatory effect of RPE-derived exosomes, they just might be used as a treatment for uveitis. It has been found that MSCs-derived exosomes from bone marrow can mitigate autoimmune uveoretinitis [124]. Since it is evident that only a few studies have examined the efficacy of MSC-secretome in uveitis, the need for additional research is unquestionable.

### 4.5. Type 1 Diabetes Mellitus

Type 1 (or insulin-dependent) diabetes mellitus (T1DM) is a chronic autoimmune disease that results in the destruction of insulin-producing β-cells in the pancreas [125]. Consequently, the reduced level of insulin finally leads to hyperglycemia. Repeated insulin injections indicate that insulin levels are not stable in affected subjects and that T1DM patients require long-lasting treatment. 

Immunotherapies and cell replacement, which include islet transplantation and stem cell differentiation into β-cells, are the primary treatment strategies for T1DM [126,127]. The simultaneous restorative and immunomodulatory properties of EVs aligned with new hypotheses regarding the nature of T1DM could make them a favorable treatment option. In general, therapies for T1DM should be able to restore B cells while modulating the immune system [128]. Autoreactive T cells have been recognized as the primary attackers due to immune dysregulation. In contrast, regulatory T cells have lost their ability to generate peripheral tolerance to β-cells [129]. EVs derived from distinct MSCs frequently exert their immunomodulatory effects in different ways.

In murine models of T1DM, EVs that originated from bone marrow-derived MSCs resulted in a delayed onset of T1DM, the preservation of islet cells, the reduction in insulitis and T cell infiltration, the decrease in antigen-presenting cell (APC) and T cell activation, lessening the Th1/Th17 population, and a decline in the production of inflammatory cytokines including IL-17, IL-6, IFNγ, TNFα, and IL-12 [124]. It was detected in mice that β(MIN6)-cell-derived EVs reduced the number of pro-inflammatory macrophages in pancreatic islets and enhanced islet angiogenesis [130]. 

EVs from human adipose tissue-derived MSCs increased the number of regulatory T cells in the spleen, raised the levels of immunomodulatory cytokines (like IL-4, IL-10, and TGFβ), and decreased the production of inflammatory cytokines such as IFNγ and IL-17 [131]. Accordingly, EV therapy stabilized blood glucose and eradicated diabetes symptoms.

The total exosome fraction isolated from Psh-Fas-anti-miR375 transfected human bone marrow-derived MSCs co-cultured with peripheral blood MSCs was found to reduce islet cell apoptosis and stimulate insulin release. Moreover, exosomes increased the number of regulatory T cells, reduced the levels of inflammatory cytokines (IL-2 and IFNγ), and diminished immune-rejection after islet transplantation [132].

It was found that human bone marrow-derived MSC-EVs induced the immature IL-10-producing phenotype of dendritic cells, increased the IL-10 and IL-6 levels, and the number of regulatory T cells, while it decreased the number of Th17 cells [133]. Last but not least, EVs derived from human cord blood stem cells induced monocyte differentiation into anti-inflammatory M2 macrophages (Figure 5) [134]. 

Despite the encouraging results, additional research is required to optimize the therapeutic administration of the MSC-secretome in T1DM.

### 4.6. Inflammatory Bowel Diseases

The term inflammatory bowel disease (IBD) refers to a group of disorders that are characterized by chronic, progressive immune-mediated inflammation of the digestive tract with frequent extra-intestinal, systemic organ involvement. Ulcerative colitis (UC) and Crohn’s disease (CD) are listed as the prototype disorders [135]. The GI-tract plays a crucial role in maintaining the host’s immune homeostasis, so a disrupted balance between immune tolerance and inflammation may lead to IBD [135]. 

Based on animal models of IBD, the use of MSCs has been shown to have effective anti-inflammatory and tissue repair actions. The efficacy and safety of the MSC-secretome have also been demonstrated in animal studies. It is hypothesized that if MSCs are injected locally into the therapy-refractory fistulas of Crohn’s disease patients, the closure of fistulas is mainly due to the immunomodulatory activity of the secretome, instead of the MSCs per se [136]. 

The local or systemic administration of MSC-derived EVs in mice IBD models induced by dextran sodium sulfate (DSS) has been shown to ameliorate the clinical severity of colitis, resulting in improved survival rates and reduced gut inflammation in histology [137,138,139]. These positive effects have been linked to the polarization of macrophages toward the M2 phenotype as well as the reduction in oxidative stress in the affected tissues [139,140]. It has also been demonstrated that IFNγ- and TNFα-preconditioned MSCs enrich EVs in HGF, PGE2, or TGFβ [137]. In mice models of DSS-induced colitis, MSC-derived conditioned medium (MSC-CM) has been found to downregulate the expression of TNFα or IL-1 but upregulate IL-10 [141].

In vitro exposure of macrophages to the MSC-derived secretome, followed by their implantation in a 2,4,6-trinitrobenzenesulfonic acid (TNBS)-induced colitis mouse model, demonstrated that preconditioned macrophages enhanced the survival rate by decreasing weight loss and bloody stool, thereby preventing colitis progression and disease recurrence [142].

Although animal and cell culture researches in IBD have indicated the favorable effect of the MSC-secretome, human studies are still needed.

### 4.7. Graft-Versus-Host Disease

Graft versus host disease (GvHD) represents a potentially fatal multiorgan complication of allogeneic bone marrow or peripheral blood HSC transplantation. It is based on the fact that the host appears to non-self to the graft and so it can stimulate antigenically, employing a broad spectrum of immune mechanisms. GvHD is classified into acute and chronic types [143]. The chronic form frequently mimics autoimmune diseases.

In 2004, it was reported that the immunomodulatory qualities of MSCs were effective in curing GvHD caused by HSCT [144]. Since then, extensive preclinical and clinical trials in humans have investigated the use of MSCs to treat GvHD. However, the outcomes are inconsistent and contentious [145,146]. This has been primarily attributed to technical issues, but it may also be due to the ability of each patient’s unique microenvironment to polarize the anti-inflammatory MSC2 phenotype. In this situation, cell-free therapies could be a suitable option. 

Undoubtedly, EVs have demonstrated promising results in the prevention and treatment of GvHD [147]. Using MSC-EVs to treat acute and chronic GvHD has been shown to reduce the severity of clinical manifestations (e.g., symptoms of liver, gastrointestinal tract, skin) [148]. Recently, in an acute GvHD mouse model, the effects of bone marrow-MSC-derived EVs administered intraperitoneally were investigated. The results underscored that EVs could not only lessen the severity of the disease but also stop it from getting worse, and so increase the number of animals that survived [149]. Despite all of this, human studies have not begun yet.

### 4.8. Psoriasis

Psoriasis refers to a chronic, immune-mediated, non-infectious inflammatory skin disease with keratinocyte hyperproliferation in the epidermis that is characterized clinically by the development of reddish, scaly papules and plaques. Several subtypes can be distinguished. Systemic manifestations such as painful (spondylo)arthritis or ocular involvement could also occur [150]. Psoriasis is considered as an autoimmune disease.

Until recently, there have been no applicable animal models of psoriasis, so the xenotransplantation of psoriatic skin pieces from human to mice with severe combined immunodeficiency was utilized [151]. The occurrence of animal models that have been genetically modified has altered this circumstance. Mice with an epidermis-specific deletion of the inhibitor of nuclear factor-B kinase 2 can develop a psoriasis-like skin disease independent of T cells [152]. It has been recently demonstrated that the subcutaneous injection of hUCMSCEs significantly suppressed the proliferation of epidermis and reduced the psoriasis area and severity index scores in imiquimod-induced mice [153]. Moreover, the expression of IL-17, IL-23, and CCL20 was reduced, and the phosphorylation of STAT3 was inhibited both in the skin of mice and human keratinocytes. In addition, the maturation and activation of dendritic cells were inhibited when co-cultured with hUCMSCEs in vitro, and the expression level of IL-23 was also reduced. Furthermore, in a rat model of psoriasis induced by imiquimod, the topical application of a cream containing bone marrow-derived MSC-secretome ameliorated the skin lesions [152]. 

In a recent and only human study [154], adipose tissue-derived MSCs were collected, multiplied, and then the MSC-CM was extracted and concentrated. The conditioned medium was topically applied to the patients’ affected skin areas once a day over one month. The clearance of silvery scales and the severity of plaques were completely abolished within the treatment period. Moreover, the beneficial effect of the treatment was observed after 6 months.

The results so far are encouraging for skin diseases, but further studies are required to determine the route and duration of application for human therapy.

### 4.9. Atopic Dermatitis

Atopic dermatitis (AD) is a common, heterogeneous, chronic, itchy, inflammatory skin disease frequently with increased serum IgE. Defects in terminal keratinocyte differentiation and altered immune responses are cardinal disease determinants. AD can be the initial manifestation of a spectrum of allergic diseases (the “atopic march” theory) [155].

It has been reported that exosomes derived from adipose tissue-derived MSCs (AT-MSCs) inhibit the lipopolysaccharide-induced nitric oxide production of RAW264.7 macrophages [155]. In a murine model of AD, the subcutaneous and intravenous administration of the AT-MSC-exosomes mitigated the symptoms and decreased the serum IgE levels in a dose-dependent manner. In AD-like lesions, the AT-MSC-exosomes reduced the number of CD86+ cells, mast cells, and CD206+ cells. After systemic administration, these exosomes alleviated AD-like symptoms, and in skin lesions, the mRNA levels of IL-4, IL-31, IL-23, and TNFα were reduced [156,157]. 

Therefore, it seems that AT-MSC-exosomes are able to favorably influence the outcome of AD, but this hypothesis demands further investigations.

### 4.10. Asthma

Asthma is a heterogeneous chronic inflammatory disease of the lungs’ airways provoked by exogenous (like air pollution, cigarette smoke) and endogenous factors (such as oxidative stress, chronic inflammation), and are mainly driven by a “type 2-high” immune signature [158]. Most current medications focus on regulating the inflammatory cascade, but cannot reverse tissue remodeling.

Some years ago, in an ovalbumin-induced murine asthma model (OVA-mice) [159], it was demonstrated that adipose-tissue derived EVs were superior to MSCs in terms of decreasing pro-inflammatory mediators and inflammatory cell infiltration. Both therapeutic interventions also reduced the number of eosinophils, collagen fiber content, and TGFβ level in a comparable manner [159]. Furthermore, in OVA-mice [160], hypoxic hUCMSCEs significantly ameliorated eosinophils and pro-inflammatory mediators (IL-4 and IL-13). Moreover, hypoxic-EVs were generally more potent in suppressing airway inflammation, preventing airway remodeling, and decreasing the expression of pro-fibrogenic markers [160]. Additionally, in OVA-mice, it was also found that EVs from MSCs in adipose tissue helped reduce allergic airway inflammation and airway hyperresponsiveness caused by the activation of regulatory T cells to expansion [161]. Despite the encouraging results, to identify the EV components responsible for the suppression of allergic airway inflammation in asthmatic mice, additional investigations are required. Moreover, human experiments provide greater biological complexity than model experiments with mice. Thus, human models are needed to figure out what kind of role adipose-tissue MSC-derived EVs might play in allergic airway inflammation.

## 5. Limitations of the Use of the MSC-Secretome

Not only would it be important to investigate the use of MSC-secretome in ADs and IMIDs because it is a relatively simple, cell-free immunomodulatory method and does not have the same ethical implications as MSC treatments, but the side effects of the MSC-secretome should also be considered. 

The factors that introduce variability in the secretome composition (e.g., the donor-related ones such as obesity, aging; the cell source-related factors; and cell passage) have been recently summarized [69]. Noting that variability introduced by the cell donor or origin renders MSCs unpredictable, there is a growing interest in replacing them with immortalized cell lines [162]. It was demonstrated that the immortalized MSC-derived secretome does not vary based on the tissue source or cell passage [163].

In addition to the technical limitations above-mentioned, it was reported that epigenetic modifications of MSCs (e.g., viral infections) also affect the composition of the MSC-secretome, and thus both MSCs and the MSC-secretome may contribute to the development, maintenance, and progression of inflammatory diseases [100,101,102]. 

However, one of the most worrying of these is the protumor effect [8]. MSCs are recruited to assist in the repair and regeneration of damaged tissues. As a result of intense cell recruitment and cross-modulation, aggressive tumor development also leads to inflammation-related tissue injury. MSCs can interact with tumor cells by exchanging secretome [164,165,166,167], encouraging reciprocal exchange, and inducing biological markers [168,169]. Aside from the direct effect of the MSC-secreted soluble fraction, enzymes excreted into and activated within exosomes (primarily MMPs and their regulators) may confer novel properties on malignant cells [164]. The vesicular fraction of the secretome is involved in pre-metastatic niche formation and tumor neovascularization. Furthermore, abnormalities in the extracellular matrix may influence the progression of cancer by promoting fibroblastic switching and mesenchymal mode acquisition [165]. Exosome incorporation has been linked to the development of ecto-5′-nucleotidase activity in some tumor cells [164]. Using the stimulation of adenosine receptor signaling located on the external membrane of the majority of immunocompetent cells (e.g., tumor-infiltrating T-cell function), tumor cells with this unique capability are capable of suppressing and modulating inflammation-inducing activity [170,171]. Using their secretome, tumor cells can also affect and modify MSCs in the opposite direction [165,172]. Extracellular vesicles secreted by cancer stem cells can set up a metastasis-supportive compartment and cause an epithelial-to-mesenchymal transition, which allows for tumor spread [165]. These undesirable aspects of MSC-secretome therapy need to be studied more in depth in any case. 

## 6. Concluding Remarks

The efficacy of conventional non-specific immunosuppressive medications for autoimmune diseases is limited, especially in severe refractory phenotypes, and could be accompanied by several harmful side effects. Therefore, constant efforts are being made to develop a new generation of highly effective treatment alternatives. 

In this review, we focused on the major biological, molecular, and pathological effects of administering the MSC-secretome to animal models or in vitro cell cultures of ADs and IMIDs as well as the human results reported to date.

A broad spectrum of MSC-secretome-related biological activities has been proven so far including anti-inflammatory, anti-apoptotic, and immunomodulatory properties, making the secretome theoretically capable of restoring aberrant immune regulation (i.e., achieving immune homeostasis). In comparison with MSCs, the highly biocompatible secretome is far less immunogenic but exerts similar biological actions, so it can be considered as an ideal cell-free therapy option. Moreover, since the composition of the MSC secretome can be engineered, from a future perspective, it could also be viewed as part of a potential delivery system within nanomedicine, allowing us to specifically target dysfunctional cells or tissues.

Although, until recently, many encouraging results from pre-clinical studies using different animal models and cell culture experiments have been obtained that strongly support the application of the cell-free MSC-secretome in various autoimmune and immune-mediated inflammatory conditions, human studies with MSC-secretome administration are still in their infancy. According to www.clinicaltrials.gov (accessed on 22 May 2022), there are currently only five ongoing human studies that deal with the therapeutic potential of the MSC-secretome in IMIDs. These include investigations that have already been completed (uveitis), are in progress (asthma), or are still being planned (T1DM, corneal defect, IBD). Therefore, much effort has to be made to translate the promising experimental data of MSC-secretome utility to daily clinical practice with a special focus on its safety. For the purpose of the challenging human applications, the development of standardized secretome isolation, purification, and storage methods/protocols, and quality assurance parameters is indispensable. Furthermore, the dose and route of secretome administration must also be clearly determined.

In summary, therapy with MSC-secretome as a cell-free alternative biological approach may expand the therapeutic horizons of autoimmune and immune-mediated inflammatory diseases.

## Figures and Tables

**Figure 1 cells-11-02300-f001:**
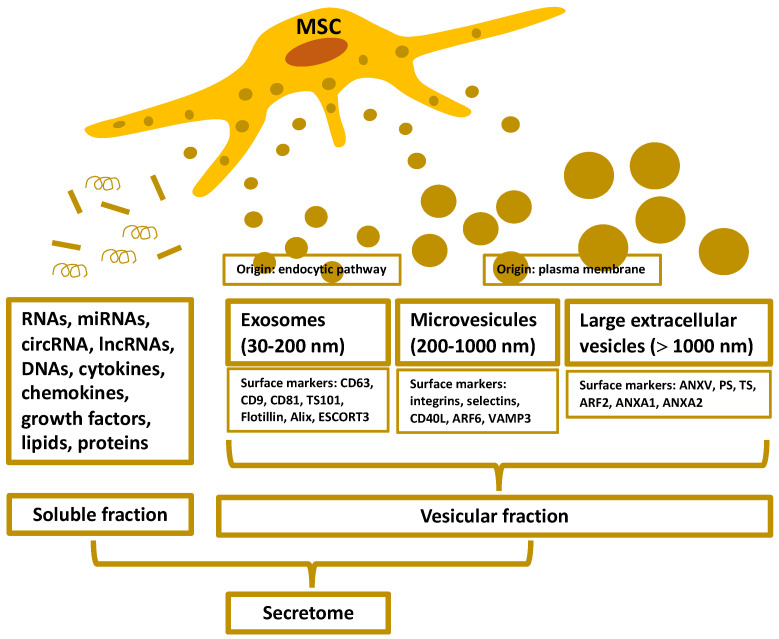
The secretome is defined as the set of substances released from the cell (e.g., mesenchymal stem cells—MSCs) to their surroundings with a wide spectrum of biological action. The figure represents the components, their origin, size, and characteristic surface markers. (RNA: ribonucleic acid; miRNA: micro-RNA; circRNA: circular-RNA; lncRNA: long non-coding-RNA; DNA: deoxyribonucleic acid; TS101: tumor susceptibility 101; ESCORT3: endosomal sorting complex required for transport-3; CD40L: CD40 ligand; ARF: ADP-ribosylation factor; VAMP3: vesicle-associated membrane protein 3; ANX: Annexin; PS: phosphatidylserine; TS: thrombospondin).

**Figure 2 cells-11-02300-f002:**
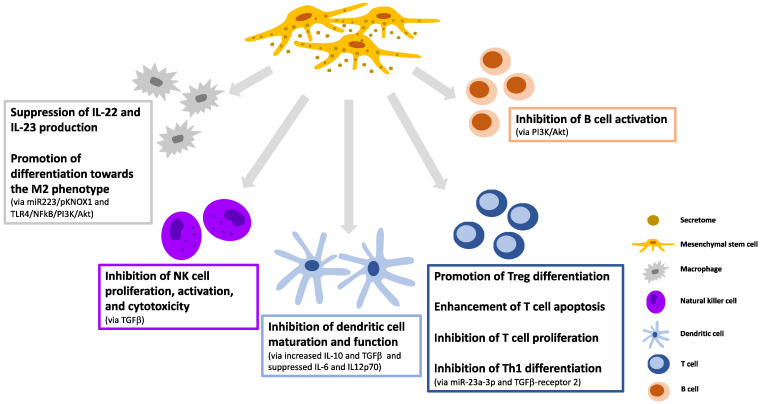
The beneficial immunomodulatory effects of the MSC-secretome. The diagram illustrates how MSC-secretome components contribute to the amelioration of inflammation. (IL: interleukin; pKNOX1: PBX/knotted 1 homeobox 1; TLR: Toll-like receptor; PI3K: phosphatidylinositol 3-kinase; NFkB: nuclear factor kB; TGF: transforming growth factor; Treg: regulatory T cell; Th1: type 1 helper T cell).

**Figure 3 cells-11-02300-f003:**
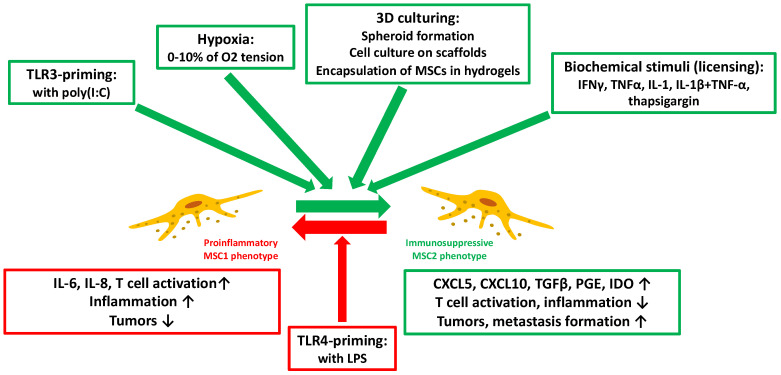
The modes and biological effects of MSC-secretome manipulation. Methods marked with green and green arrows favor the appearance of the MSC2 immunosuppressive phenotype. The method marked in red and the red arrow promote the MSC1 proinflammatory phenotype (TLR: Toll-like receptor; poly (I:C): polyinosinic:polycytidylic acid; 3D: 3 dimensional; MSC: mesenchymal stem cell; IFN: interferon; TNF: tumor necrosis factor; IL: interleukin; LPS: lipopolysaccharide; CXCL: C-X-C motif chemokine ligand; TGF: transforming growth factor; PGE: prostaglandin E; IDO: indolamin-2,3-dioxigenase).

**Figure 4 cells-11-02300-f004:**
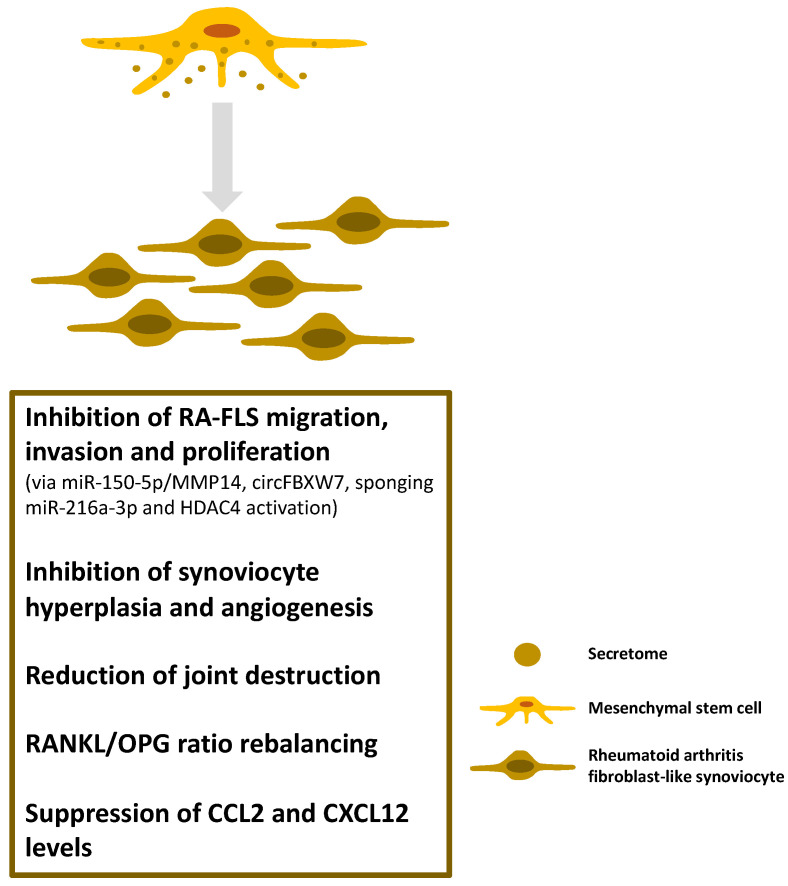
The anti-inflammatory and immunomodulatory effects of the MSC-secretome in rheumatoid arthritis. The figure depicts how components of the MSC-secretome contribute to the amelioration of disease activity. (RA-FLS: rheumatoid arthritis fibroblast-like synoviocyte; MMP: matrix metalloproteinase; HDAC4: histone deacetylase 4; RANKL: receptor activator of nuclear factor kappa-B ligand; OPG: osteoprotegerin; CCL2: C–C motif chemokine ligand 2; CXCL12: C-X-C motif chemokine ligand 12).

**Figure 5 cells-11-02300-f005:**
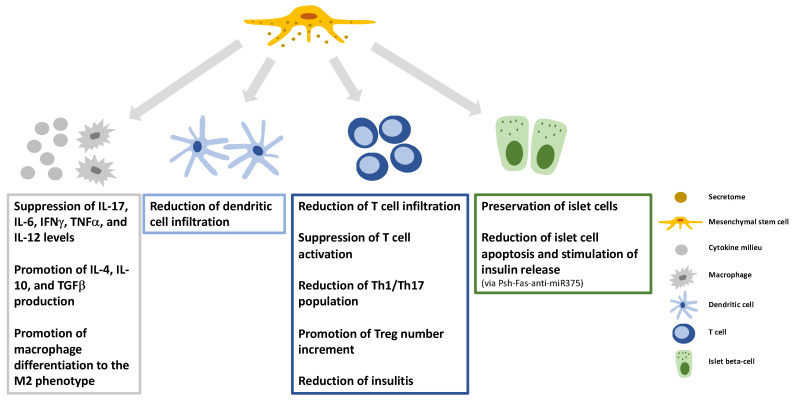
The potential modes of beneficial action of the MSC-secretome in type 1 diabetes mellitus (T1DM). The figure shows how MSC-secretome contributes to the restoration of immune dysregulation in T1DM (IL: interleukin; TNF: tumor necrosis factor; IFN: interferon; TGF: transforming growth factor; Treg: regulatory T cell; Th1/Th17: type 1/type 17 helper T cell).

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
