# Peer review of "Mesenchymal Stem Cell-Derived Secretome: A Potential Therapeutic Option for Autoimmune and Immune-Mediated Inflammatory Diseases"

_cells, 2022, doi:10.3390/cells11152300_

Round 1

Reviewer 1 Report

Györgyi Műzes and Ferenc Sipos have summarized the information on the immunomodulatory effects of the MSC-secretome in some of the most relevant autoimmune-mediated inflammatory diseases with special emphasis on its potential as therapeutic tool. In addition to describing the basic concepts on the composition of the MSC secretome and factors that can modulate it, this review also includes specific evidence, collected until now mostly in animal models, of its therapeutic application in different autoimmune diseases.

I hope the following comments will help them to improve the manuscript.

1. This is a correctly structured review paper in which a logical organization of the content can be appreciated considering the titles of the sections in which it is organised. However, there are some sections missing, or at least a paragraph referring to relevant issues, such as: technique for EV isolation,

2. Although some comments on the limitations of the therapeutic use of the secretome are made in the introductory and concluding sections, these are very superficial. It would be interesting to include an analysis of the actual specific limitations involved in the use of the secretome, referring to delivery models, the variability associated with the cell tissue source or cell passages and also outlining the potential solutions that have emerged to address these shortcomings, such as the use of immotalised MSC lines.

3. Figure 1: This figure could be much more informative if it were completed with a small table describing the basic characteristics of the three types of EVs, i.e, biogenesis mechanism, size and surface markers.

4. Line 111-112: “By preconditioning MSCs, which promotes the expression of the MSC2 phenotype, the immunomodulatory effect of MSC-secretome can be markedly enhanced”. This sentence should be repositioned in the text as it provides information on MSC2 without first describing its characteristics. It is confusing and does not follow a logical structure in the content of the text.

5. Section 3: A few lines could be introduced at the beginning to explain the concepts of polarisation and plasticity.

6. The paragraphs between lines 185 and 198 contain information regarding the immunomodulatory effects of the MSC secretome but these are not the results of experiments in either sle patients or murine models. These paragraphs describe general processes applicable to all diseases with an autoimmune inflammatory profile, so including them in the SLE-specific subsection is somewhat confusing. A general section on immunomodulatory activity of the secretome could be considered.

7. Lines 266-270: The wording of these lines is confusing and difficult to understand. The writing should be revised.

Author Response

Dear Reviewer 1,

We appreciate your constructive criticism, helpful advice, and insightful comments.

Our answers to your comments are summarized below.

# 1. „This is a correctly structured review paper in which a logical organization of the content can be appreciated considering the titles of the sections in which it is organised. However, there are some sections missing, or at least a paragraph referring to relevant issues, such as: technique for EV isolation,”

Our answer: We have inserted a brief paragraph on the isolation techniques of secretome after the description of the organization of EVs.

# 2. „ Although some comments on the limitations of the therapeutic use of the secretome are made in the introductory and concluding sections, these are very superficial. It would be interesting to include an analysis of the actual specific limitations involved in the use of the secretome, referring to delivery models, the variability associated with the cell tissue source or cell passages and also outlining the potential solutions that have emerged to address these shortcomings, such as the use of immotalised MSC lines.”

Our answer: We have also inserted a new paragraph regarding the potential side effects of secretome therapy, including the suggested aspects.

# 3. „Figure 1: This figure could be much more informative if it were completed with a small table describing the basic characteristics of the three types of EVs, i.e, biogenesis mechanism, size and surface markers.”

Our answer: Figure 1 has been restructured according to your suggestions.

# 4. „Line 111-112: “By preconditioning MSCs, which promotes the expression of the MSC2 phenotype, the immunomodulatory effect of MSC-secretome can be markedly enhanced”. This sentence should be repositioned in the text as it provides information on MSC2 without first describing its characteristics. It is confusing and does not follow a logical structure in the content of the text.”

Our answer: This sentence has been deleted since, as you mentioned, this sentence referred prematurely to something we will explain later.

# 5. „Section 3: A few lines could be introduced at the beginning to explain the concepts of polarisation and plasticity.”

Our answer: In between lines 114-124 (original numbering), we wrote about these suggested aspects.

# 6. „The paragraphs between lines 185 and 198 contain information regarding the immunomodulatory effects of the MSC secretome but these are not the results of experiments in either sle patients or murine models. These paragraphs describe general processes applicable to all diseases with an autoimmune inflammatory profile, so including them in the SLE-specific subsection is somewhat confusing. A general section on immunomodulatory activity of the secretome could be considered.”

Our answer: According to your suggestion, we replaced the general immunomodulatory functions in the first part of Section 3, and we also highlighted and supplemented the SLE part with research data on the disease itself. This also resulted in Figure 2 having been replaced and its legend having been rephrased.

# 7. „ Lines 266-270: The wording of these lines is confusing and difficult to understand. The writing should be revised.”

Our answer: The indicated sentences have been rephrased for better understanding.

All the changes you suggested are highlighted in purple in the manuscript.

We really hope that you agree with the changes you suggested for the paper and the answers to your questions, and that this revised article is now ready to be published.

Reviewer 2 Report

The authors took on the task of reviewing the current status of the clinical application of MSC secretome in Immune-mediated inflammatory diseases (IMIDs). It’s an interesting and important issue to be addressed and the authors made an effort to design a adequate review. 

There are, however, several issues that need to be addressed before I recommend publication of the manuscript. 

1)    An overview of the current status of the clinical use of MSC secretome in diseases other that the autoimmune disorders is expected and the authors need to address it in a short but comprehensive paragraph

2)    The paragraph that talks about the clinical use of stem cells needs to distinguish between stem cells and mesenchymal stem cells emphasizing in the specific characteristic of the latter (paragraph that starts with “Adult stem cells (SCs) such hematopoietic stem cells and mesenchymal stem cells (MSCs) are by far…”, lines 50-64. ) 

3)    Better description of the distinct characteristics between oncosomes and apoptotic bodies should be provided. 

4)    Section 3 on “Immunomodulatory effects and manipulation of the MSC-secretome” should be expanded and authors should also provide a figure with a summary of current status of ways of manipulation of the MSCs’ sectretome. 

5)    In section 5 “Concluding remarks” the first paragraph is a repetition of the introduction. An effort should be made in order to provide useful conclusions regarding of the use of the secretome in the different diseases that the authors provide information for. 

Minor issues:

1)    Lines 139-140 is not understandable what the authors want to say

2)    Line 102 the “whole heterogenous messenger substances” needs to be rephrased.

Author Response

Dear Reviewer 2,

We value your constructive feedback, helpful suggestions, and insightful remarks.

Our responses to your remarks are outlined below.

# 1. „ An overview of the current status of the clinical use of MSC secretome in diseases other that the autoimmune disorders is expected and the authors need to address it in a short but comprehensive paragraph”

Our answer: The current clinical and research applications of MSC-secretome have been summarized.

# 2. „ The paragraph that talks about the clinical use of stem cells needs to distinguish between stem cells and mesenchymal stem cells emphasizing in the specific characteristic of the latter (paragraph that starts with “Adult stem cells (SCs) such hematopoietic stem cells and mesenchymal stem cells (MSCs) are by far…”, lines 50-64. )”

Our answer: We have extended the characterization of MSCs.

# 3. „Better description of the distinct characteristics between oncosomes and apoptotic bodies should be provided.”

Our answer: A detailed description has been added to the text.

# 4. „Section 3 on “Immunomodulatory effects and manipulation of the MSC-secretome” should be expanded and authors should also provide a figure with a summary of current status of ways of manipulation of the MSCs’ sectretome.”

Our answer: A general chapter about the immunomodulatory effects of MSC-secretome has been inserted into the text. Also, a new figure has been created.

# 5. „ In section 5 “Concluding remarks” the first paragraph is a repetition of the introduction. An effort should be made in order to provide useful conclusions regarding of the use of the secretome in the different diseases that the authors provide information for.”

Our answer: As suggested by you, we have briefly summarized the essence of the article. However, we have already summarized the main effects of the MSC secretome, referred to the possibilities of modifying the composition of the secretome, and emphasized that despite the promising in vitro and animal model results, there is little human experience. We have also already emphasized that more human research is needed and special attention should be paid to the safety of secretome therapy. At the same time, it is not our intention to highlight individual diseases again in the Conclusion section, as the article itself addresses this.

# 6. „ Lines 139-140 is not understandable what the authors want to say.; Line 102 the “whole heterogenous messenger substances” needs to be rephrased.”

Our answer: Lines 139-140 and 102 have been rephrased.

All suggested revisions are highlighted in green throughout the manuscript.

We sincerely hope that you accept the suggested corrections to the paper and that this revised article is now suitable for publication.

Reviewer 3 Report

The authors review the application of mesenchymal stem cell secretome as cell-free therapy in the most common autoimmune diseases.

Minor Concerns:

1-The authors write in the abstract "MSC-secretome (containing extracellular vesicles and exosomes)" (line 13). Exosomes are extracellular vesicles so mentioning them separately is a mistake. This sentence needs to be corrected.

2-The authors reference Sipos on line 62 to mention the dangers of mesenchymal stem cell therapies, however Sipos' article deals with the controversy in therapeutic application of mesenchymal stem cell-derived secretome.

3-On line 102, in the legend, secretoma is misspelled.

4- On line 539 the authors reference de Castro LL's article as recently published, however, this article is from 2017. Can the authors substitute the word recent in the manuscript?

Author Response

Dear Reviewer 3,

We value your constructive criticism, beneficial suggestions, and insightful comments.

The following is a summary of our responses to your remarks.

# 1. “The authors write in the abstract "MSC-secretome (containing extracellular vesicles and exosomes)" (line 13). Exosomes are extracellular vesicles so mentioning them separately is a mistake. This sentence needs to be corrected.”

Our answer: The sentence has been corrected.

# 2. „The authors reference Sipos on line 62 to mention the dangers of mesenchymal stem cell therapies, however Sipos' article deals with the controversy in therapeutic application of mesenchymal stem cell-derived secretome.”

Our answer: The title of the article refers to the controversies of MSC-secretome therapy, but one chapter of the article deals with the pros and cons of MSC therapies.

# 3. „On line 102, in the legend, secretoma is misspelled.

Our answer: The misspelled word has been corrected.

# 4. “On line 539 the authors reference de Castro LL's article as recently published, however, this article is from 2017. Can the authors substitute the word recent in the manuscript?”

Our answer: Recently was substituted by „some years ago”.

All suggested revisions are highlighted in blue throughout the manuscript.

We sincerely hope that you accept the recommended modifications to the paper and the answers to your questions, and that this revised article is now ready for publication.

Reviewer 4 Report

The article entitled ” Mesenchymal stem cell-derived secretome: a potential therapeutic option for autoimmune and immune-mediated inflammatory diseases” reviewed studies of immunomodulatory effects of MSC-derived EVs in different inflammatory diseases, suggesting that it is an ideal cell-free therapeutic alternative compared to standard pharmacological therapies and MSC themselves. However, this review is poorly written, and scientifically unsound, it lacks a concise presentation of the problems in the field and critical discussion on the data presented, especially when in comes to translation of the results obtained from xenogenic model systems. It is not clear why the review was necessary when similar papers were published already(i.e. https://www.frontiersin.org/articles/10.3389/fimmu.2021.749192/full). Too many reviews were cited without referring to original papers. Additionally, some citations were misinterpreted (i.e. 47-49) or wrongly used (i.e. 96), and many sentences are logically incomplete, scientifically unsound or wrong i.e. sentences in lines 119-120, 139-140, 142, 147, 153-155, 168, 196, 197, 215, etc…). Importantly, there is a lack of discussion related to the authors' own research in this area.

Author Response

Dear Reviewer 4,

We appreciate your constructive criticism.

Our responses to your remarks are outlined below.

# 1. “this review is poorly written, and scientifically unsound, it lacks a concise presentation of the problems in the field and critical discussion on the data presented, especially when in comes to translation of the results obtained from xenogenic model systems. It is not clear why the review was necessary when similar papers were published already(i.e. https://www.frontiersin.org/articles/10.3389/fimmu.2021.749192/full).”

Our answer: In the review, we have emphasized issues that have not yet been addressed. We have attempted to compile these in such a way as to display all the data and results regarding autoimmune diseases that have never been discussed in any previous communication. The structure of our article, as well as the communication's logic, was designed primarily for immunologists with clinical experience, taking into account aspects of daily practice and presenting future possibilities. We did not wish to delve into technical details, as numerous studies have already been published on these topics.

You write that similar articles have been written on the subject. The comment itself includes the answer to why we wrote the article. Compared to the article you mentioned, our article is new based on the following criteria:

  1. Regarding the innate and adaptive immune systems, the said article confuses innate and adaptive immune system effects in IMID and non-IMID diseases. Furthermore, it does not distinguish between inflammatory and non-inflammatory milieus. We, on the other hand, don't mix up the effects of MSC-secretome on the innate and adaptive immune systems. Instead, we focus on diseases that are definitely autoimmune or IMID, since the inflammatory medium alone can also change the effects of MSC-secretome.
  2. The previous article discusses a total of six diseases but does not discuss the results of research on SpA, GvHD, psoriasis, atopic dermatitis, and asthma. Also, that article doesn't talk about the most recent findings, while our article does.

# 2. „Too many reviews were cited without referring to original papers.”

Our answer: Essentially, we only cite the original articles. In some instances, we make an exception and cite review articles because they include relevant original communications and contain information that we do not wish to include in detail.

# 3. “Additionally, some citations were misinterpreted (i.e. 47-49) or wrongly used (i.e. 96)”

Our answer: Regarding references (originally No. 47 and 48), the structure of the immunomodulatory effects of MSC-secretome has been revised. Now, the interpretation of the references is correct. Reference No. 96 has been deleted.

# 4. “many sentences are logically incomplete, scientifically unsound or wrong i.e. sentences in lines 119-120, 139-140, 142, 147, 153-155, 168, 196, 197, 215, etc…).

Our answer: Lines 119-120, 142, 168, 197, and 215 are all understandable phrases or clinical terminologies. Line 139-140: „fate" has been corrected to "death". Lines 147, 153, and 155 have been corrected.

# 5. „Importantly, there is a lack of discussion related to the authors' own research in this area.”

Our answer: It is not a prerequisite for a review article that the authors have research results on a particular topic. The article was written from the clinical immunologist's point of view. In addition, we also dealt with stem cells on another topic at the research level.

All suggested revisions are highlighted in yellow throughout the manuscript.

We sincerely hope that you accept the suggested corrections to the paper and the answers to your questions, and that this revised article is now acceptable for publication.

Round 2

Reviewer 2 Report

I still have an issue regarding the classification of adult stem cells and characterisation of MSCs (point No2 in the previous round of revision). A recent article considering the classification of stem cells should be used. 

Author Response

Dear Reviewer 2,

The classification and characterization of stem cells are further detailed, according to your suggestion.

Two new references have also been added:

1., Shah AA, Khan FA. Types and Classification of Stem Cells. In: Advances in Application of Stem Cells: From Bench to Clinics. Editor: Firdos Alam Khan 1st ed.; Humana Press: Totowa, New Jersey, USA, 2021, pp. 25-49.

2., Falzarano MS, Ferlini A. Urinary Stem Cells as Tools to Study Genetic Disease: Overview of the Literature. J Clin Med. 2019;8:627.

Yours Sincerely,

Műzes and F. Sipos

Reviewer 4 Report

My major concern about this review was that it does not provide the author's own critical perspective and original contribution to this field. Although the authors changed somewhat these issues, the major problem still holds true. For example, the authors stated that "TLR3 activation with inflammatory cytokines like IFNγ, TNFα, or IL-1 polarizes MSCs toward the MSC2 anti-inflammatory phenotype, producing...." and then cited a review article 67.  TLR3 is an endosomal receptor recognizing double-stranded RNAs or Poly(I:C) as a synthetic agonist. Therefore it is not true that these cytokines activate TLR3. This is only one of the mistakes made by authors, so I am still not convinced that this is an acceptable article that would make a significant contribution to the field, quite contrary.

Author Response

Dear Reviewer 4,

Our responses to your remarks are outlined below.

  1. “My major concern about this review was that it does not provide the (1.) author's own critical perspective and (2.) original contribution to this field.”
  2. “the authors stated that "TLR3 activation with inflammatory cytokines like IFNγ, TNFα, or IL-1 polarizes MSCs toward the MSC2 anti-inflammatory phenotype, producing...." and then cited a review article 67. TLR3 is an endosomal receptor recognizing double-stranded RNAs or Poly(I:C) as a synthetic agonist. (3.) Therefore it is not true that these cytokines activate TLR3.”

Our responses:

Ad 1.

In autoimmunity, the therapeutic use of the MSC-secretome is still in its infancy. Unfortunately, very few human studies are conducted and very few human results are available. On the basis of these, it would be extremely challenging for anyone to formulate a pointedly critical comment or opinion on the topic. In addition, we do not wish to criticize the use of the MSC-secretome but rather to encourage the expansion of clinical research in this area. Moreover, we emphasized this throughout the article (see the list below).

At the same time, we treat the results critically, as we are not only highlighting the technical and cost issues, but it has been nearly impossible to find an article that discusses one of the most dangerous potential side effects of this treatment, namely tumorigenesis. We also do that (see list below).

Hereby, we would like to list the lines that formulate our critical opinion about each chapter (the line numberings are valid for the first revision of the manuscript):

Lines 147-149. “Clinical applications have a great need for EV isolation technologies that allow for high throughput, preserve EV integrity and biological properties, and are not prohibitively expensive.”

Lines 201-203. “The possibility of manipulating the microenvironment of MSCs to polarize them toward an MSC2 phenotype that produces a secretome rich in anti-inflammatory factors seems to be currently fashionable.”

Lines 261-263. “The theoretical basis for using MSC-secretome in SLE is partly due to MSC limitations and partly because of the fact that the secretome could affect almost all types of immunocompetent cells.”

Lines 278-280. “application of the MSC-secretome could be considered as an ideal therapeutic option for severe, refractory SLE. Additionally, MSC-secretome may represent a cell-free, low-immunogenicity alternate solution to disadvantaged MSC therapies.”

Lines 287-291. “There are numerous conventional and novel disease-modifying anti-rheumatic drugs available for the treatment of RA. Along with their immunosuppressive and immunomodulatory effects, however, several adverse effects must also be considered. Thus, the secretome derived from MSCs with immunomodulatory properties could serve as a suitable therapeutic option.”

Lines 317-323. “Although animal experiments have been extremely encouraging and no significant adverse effects associated with the use of secretome have been reported, human trials have not yet begun. It should also be borne in mind that some new research suggests that epigenetically modified bone-marrow-derived MSCs (and presumably their secretome, as well) producing more IL-6 may promote the progression of RA by migrating to the synovium. In any case, more research is required to investigate MSC-based and cell-free therapeutic options for human RA.”

Lines 332-333. “the use of MSCs and MSC-secretome in the SpA group can cause not only beneficial but also detrimental effects”

Lines 347-350. “In spondyloarthritides, the study of positive or negative therapeutic effects of the MSC-secretome is essential. In addition to animal models and cell culture experiments, detailed human studies must be introduced.”

Line 364. “The success of using MSCs in MS has fallen short of expectations.”

Lines 393-396. “Thus, exosomes secreted by microglia during virus infection could contribute to viral persistence and neuroinflammation, both of which contribute to the development of demyelinating disease.”

Lines 427-428. “Although these procedures may represent a paradigm shift in the treatment of MS, their clinical utility remains unknown.”

Lines 433-437. “Regrettably, additional evaluations are necessary to progress toward a clinically applicable functional approach. According to www.clinicaltrials.gov only one study is planned in relapsing MS, in which -as a sub-target- the analysis of molecular and biochemical parameters including circulating exosomes are planned to perform comparing responder and non-responder patients.”

Lines 468-470. “Since it is evident that only few studies have examined the efficacy of MSC-secretome in uveitis, the need for additional research is unquestionable.”

Lines 509-510. “In spite of the encouraging results, additional research is required to optimize the therapeutic administration of MSC-secretome in T1DM.”

Lines 543-544. “Although animal and cell culture researches in IBD indicate the favorable effect of MSC-secretome, human studies are still needed.”

Lines 554-558. “However, the outcomes are inconsistent and contentious. This has been primarily attributed to technical issues, but it may also be due to the ability of each patient's unique microenvironment to polarize the anti-inflammatory MSC2 phenotype. In this situation, cell-free therapies could be a suitable option.”

Lines 559-560, 565-566. “Undoubtedly, EVs have demonstrated promising results in the prevention and treatment of GvHD… Despite all this, human studies have not yet begun.”

Lines 593-594. “The results so far are encouraging for skin diseases, but further studies are required to determine the route and duration of application for human therapy.”

Lines 610-611. “So, it seems that AT-MSC-exosomes are able to favorably influence the outcome of AD, but this hypothesis demands further investigations.”

Lines 630-634. “Despite the encouraging results, for identifying the EV components responsible for the suppression of allergic airway inflammation in asthmatic mice, additional investigations are required. Moreover, human experiments provide greater biological complexity than model experiments with mice. Thus, human models are needed to figure out what kind of role might adipose-tissue MSC-derived EVs play in allergic airway inflammation.”

Lines 636-671. The whole 5.0 chapter expresses our opinion. And here we are not only discussing technical problems, we are also talking about a very little-studied but important side effect, namely tumorigenesis.

Lines 673-706. The whole Concluding remark chapter expresses our opinion. 

Ad 2.

In accordance with MDPI's official guidelines, the review article must adhere to the following:

„Review articles provide concise and precise updates on the recent progress in a given area of research. […] Review articles should be comprehensive and submitted by authors who are in the field.”

There are numerous review articles published in Cells whose authors did not conduct research in the given area. Here are some recent examples of articles (and we could continue this list with additional articles):

  1. Crucial Role of Oncogenic KRAS Mutations in Apoptosis and Autophagy Regulation: Therapeutic Implications

by Anabela Ferreira, Flávia Pereira, Celso Reis, Maria José Oliveira, Maria João Sousa and Ana Preto

Cells 2022, 11(14), 2183; https://doi.org/10.3390/cells11142183

(Only one own review is cited by the authors: Cazzanelli, G.; Pereira, F.; Alves, S.; Francisco, R.; Azevedo, L.; Dias Carvalho, P.; Almeida, A.; Côrte-Real, M.; Oliveira, M.J.; Lucas, C.; et al. The Yeast Saccharomyces Cerevisiae as a Model for Understanding RAS Proteins and Their Role in Human Tumorigenesis. Cells 2018, 7, 14.)

  1. MiRNAs as Promising Translational Strategies for Neuronal Repair and Regeneration in Spinal Cord Injury

by Serena Silvestro and Emanuela Mazzon

Cells 2022, 11(14), 2177; https://doi.org/10.3390/cells11142177

(No own contribution to the field is cited)

  1. Role of Circular RNA in Brain Tumor Development

by Swalih P. Ahmed, Javier S. Castresana and Mehdi H. Shahi

Cells 2022, 11(14), 2130; https://doi.org/10.3390/cells11142130

(No own contribution to the field is cited)

  1. Microglia Phenotypes in Aging and Neurodegenerative Diseases

by Menbere Y. Wendimu and Shelley B. Hooks

Cells 2022, 11(13), 2091; https://doi.org/10.3390/cells11132091

(No own contribution to the field is cited)

According to this, the publication of a review article does not necessitate that the authors have their own relevant research results. As physicians who treat immunological patients, our primary goal in writing this article was to emphasize the real clinical need for new therapies and a potential method for developing them.

On the other hand, we have our own research findings on the regenerative role of mesenchymal stem cells in inflammation and the appearance of the stem cell phenotype in colon cancer cells. However, these results of ours are not relevant to the current review article, so we do not quote them. We will not unethically cite our own works in order to increase the number of self-citations.

In addition, based on our previous work and ongoing research (concerning the MSC-secretome), we feel qualified to write a review article on this subject.

Furthermore, based on our prior research and publication activities as authors, Cells asked us to serve as reviewers and guest editors, which was also approved by the journal's current Editorial Board.

In light of these facts, the assumption of Reviewer 4 is even more incomprehensible and offensive.

For your information, we list our original contribution to the field:

1., Valcz G, Krenács T, Sipos F, Wichmann B, Tóth K, Leiszter K, Balogh Z, Csizmadia A, Hagymási K, Műzes G, Masszi T, Molnár B, Tulassay Z. Appearing of bone marrow derived stem cells in healthy and regenerating colonic epithelium. Orv Hetil. 2009 Oct 4;150(40):1852-7. doi: 10.1556/OH.2009.28719.

(Local stem cells probably have enough regeneration capacity in case of minor colonic inflammation. However, in aspecific inflammation the number of MSCs contributing to epithelial regeneration was elevated, suggesting their facilitated contribution to the repair process with less probable forming of local stem cell progeny.)

2., Valcz G, Sipos F, Krenács T, Molnár J, Patai AV, Leiszter K, Tóth K, Solymosi N, Galamb O, Molnár B, Tulassay Z. Elevated osteopontin expression and proliferative/apoptotic ratio in the colorectal adenoma-dysplasia-carcinoma sequence. Pathol Oncol Res. 2010 Dec;16(4):541-5. doi: 10.1007/s12253-010-9260-z. PMID: 20349162.

(The significantly elevated osteoprotegerin protein levels we found during normal epithelium to carcinoma progression may contribute to the increased fibroblast-myofibroblast transition determining stem cell niche in colorectal cancer.)

3., Valcz G, Krenács T, Sipos F, Patai AV, Wichmann B, Leiszter K, Tóth K, Balogh Z, Csizmadia A, Hagymási K, Masszi T, Molnár B, Tulassay Z. Lymphoid aggregates may contribute to the migration and epithelial commitment of bone marrow-derived cells in colonic mucosa. J Clin Pathol. 2011 Sep;64(9):771-5. doi: 10.1136/jclinpath-2011-200022. PMID: 21653659.

(Elevated number of intraepithelial CD45-BM-derived stem cells (BMDCs) at lymphoid aggregates suggests that BMDCs play a role in epithelial regeneration and that lymphoid aggregates serve as their migration route.)

4., Sipos F, Constantinovits M, Valcz G, Tulassay Z, Műzes G. Association of hepatocyte-derived growth factor receptor/caudal type homeobox 2 co-expression with mucosal regeneration in active ulcerative colitis. World J Gastroenterol. 2015 Jul 28;21(28):8569-79. doi: 10.3748/wjg.v21.i28.8569. PMID: 26229399; PMCID: PMC4515838.

(In active UC, a portion of circulating HGFR/CD133-expressing cells are committed to the epithelial lineage (CDX2, Lgr5), and may participate in mucosal regeneration by undergoing mesenchymal-to-epithelial transition.)

5., Sipos F, Bohusné Barta B, Simon Á, Nagy L, Dankó T, Raffay RE, PetÅ‘vári G, Zsiros V, Wichmann B, Sebestyén A, Műzes G. Survival of HT29 Cancer Cells Is Affected by IGF1R Inhibition via Modulation of Self-DNA-Triggered TLR9 Signaling and the Autophagy Response. Pathol Oncol Res. 2022 May 16;28:1610322. doi: 10.3389/pore.2022.1610322. PMID: 35651701; PMCID: PMC9148969.

(Autophagy, induced by different combinations of self-DNA and inhibitors is not sufficient to rescue HT29 cells from death but results in the survival of some CD133-positive stem-like HT29 cells.)

And these are our review articles dealing with the role of stem cells in regeneration:

1., Műzes G, Bohusné Barta B, Sipos F. Colitis and Colorectal Carcinogenesis: The Focus on Isolated Lymphoid Follicles. Biomedicines. 2022 Jan 21;10(2):226. doi: 10.3390/biomedicines10020226. PMID: 35203436; PMCID: PMC8869724.

2., Sipos F, Műzes G. Teduglutide-induced stem cell function in intestinal repair. J Invest Surg. 2018 Jun;31(3):253-255. doi: 10.1080/08941939.2017.1300715. PMID: 28590166.

3., Műzes G, Sipos F. Metastatic Cell Dormancy and Re-activation: An Overview on Series of Molecular Events Critical for Cancer Relapse. Anticancer Agents Med Chem. 2017;17(4):472-482. doi: 10.2174/1871520616666160901145857. PMID: 27592547.

4., Műzes G, Sipos F. Heterogeneity of Stem Cells: A Brief Overview. Methods Mol Biol. 2016;1516:1-12. doi: 10.1007/7651_2016_345. PMID: 27044045.

5., Sipos F, Műzes G. Injury-associated reacquiring of intestinal stem cell function. World J Gastroenterol. 2015 Feb 21;21(7):2005-10. doi: 10.3748/wjg.v21.i7.2005. PMID: 25717233; PMCID: PMC4326135.

6., Sipos F, Valcz G, Molnár B. Physiological and pathological role of local and immigrating colonic stem cells. World J Gastroenterol. 2012 Jan 28;18(4):295-301. doi: 10.3748/wjg.v18.i4.295. PMID: 22294835; PMCID: PMC3261524.

7., Sipos F, Leiszter K, Tulassay Z. Effect of ageing on colonic mucosal regeneration. World J Gastroenterol. 2011 Jul 7;17(25):2981-6. doi: 10.3748/wjg.v17.i25.2981. PMID: 21799643; PMCID: PMC3132248.

8., Valcz G, Krenács T, Sipos F, Leiszter K, Tóth K, Balogh Z, Csizmadia A, Műzes G, Molnár B, Tulassay Z. The role of the bone marrow derived mesenchymal stem cells in colonic epithelial regeneration. Pathol Oncol Res. 2011 Mar;17(1):11-6. doi: 10.1007/s12253-010-9262-x. PMID: 20405350.

9., Sipos F, Műzes G, Galamb O, Spisák S, Krenács T, Tóth K, Tulassay Z, Molnár B. The possible role of isolated lymphoid follicles in colonic mucosal repair. Pathol Oncol Res. 2010 Mar;16(1):11-8. doi: 10.1007/s12253-009-9181-x. PMID: 19557549.

10., Hagymási K, Molnár B, Sipos F, Galamb O, Tulassay Z. Stem cell theory of colorectal cancer and its connection with molecular-biological data. Orv Hetil. 2007 Apr 29;148(17):779-85. doi: 10.1556/OH.2007.28071. PMID: 17452307.

11., Sipos F, Tihanyi E, Molnár B, Tulassay Z. Isolation and therapeutic use of human stem cells. Orv Hetil. 2005 Jun 19;146(25):1327-33. PMID: 16106755.

12., Galamb O, Molnár B, Sipos F, Tulassay Z. Possibilities of investigation and clinical application of adult human stem cells. Orv Hetil. 2003 Nov 16;144(46):2263-70. PMID: 14702921.

Ad 3.

The sentence objected to by Reviewer 4 has been modified for better understanding (highlighted by grey color)

Note on his/her statement “Therefore it is not true that these cytokines activate TLR3”:

this is false, since in the presence of inflammatory cytokines (IFNγ, TNFα, IL1β) TLR3 is upregulated, along with the functional consequences.

1., Raicevic G, Najar M, Stamatopoulos B, De Bruyn C, Meuleman N, Bron D, Toungouz M, Lagneaux L. The source of human mesenchymal stromal cells influences their TLR profile as well as their functional properties. Cell Immunol. 2011;270(2):207-16. doi: 10.1016/j.cellimm.2011.05.010. PMID: 21700275.

2., Raicevic G, Rouas R, Najar M, Stordeur P, Boufker HI, Bron D, Martiat P, Goldman M, Nevessignsky MT, Lagneaux L. Inflammation modifies the pattern and the function of Toll-like receptors expressed by human mesenchymal stromal cells. Hum Immunol. 2010 Mar;71(3):235-44. doi: 10.1016/j.humimm.2009.12.005. PMID: 20034529. 

Yours Sincerely,

G. Műzes and F. Sipos
